# Dysosmia Is a Predictor of Motor Function and Quality of Life in Patients with Parkinson’s Disease

**DOI:** 10.3390/jpm12050754

**Published:** 2022-05-06

**Authors:** Chia-Yen Lin, Ting-Ya Chang, Ming-Hong Chang

**Affiliations:** 1Department of Neurology, Neurological Institute, Taichung Veterans General Hospital, No. 1650, Taiwan Boulevard, Sect. 4, Taichung 40705, Taiwan; linchiayen666@gmail.com (C.-Y.L.); changtingya@gmail.com (T.-Y.C.); 2Department of Post-Baccalaureate Medicine, College of Medicine, National Chung Hsing University, No. 145, XingDa Road, South Dist., Taichung 40227, Taiwan

**Keywords:** olfaction disorders, Parkinson disease, motor activity, activities of daily living, quality of life

## Abstract

(1) Background: The correlation between dysosmia with quality of life (QoL) in patients with PD was rarely reported. The study aimed to examine the effect of dysosmia on motor function and QoL in PD. (2) Methods: This cross-sectional study, performed between October 2016 and February 2021, recorded the traditional Chinese version of the University of Pennsylvania Smell Identification Test (UPSIT), the Montreal Cognitive Assessment (MoCA), the Movement Disorder Society-sponsored revision of the Unified Parkinson’s Disease Rating Scale (MDS UPDRS), and the 39-item Parkinson’s Disease Questionnaire (PDQ-39) in patients with PD. UPSIT = 19 was applied to separate the total anosmia and non-anosmia groups. (3) Results: 243 patients with PD were recruited. The total anosmia group had higher MDS UPDRS total, part II, and part III scores than the non-anosmia group. They also had worse scores on the dimensions of activities of daily living (ADL) and cognition of the PDQ-39 than the non-anosmia group. The UPSIT score correlated MDS UPDRS part III score (*p* < 0.0001), PDQ-39 ADL quartile (*p* = 0.0202), and Dopamine transporter scan (*p* = 0.0082) in the linear regression. (4) Conclusions: Dysosmia in PD predicted a phenotype with defective motor function, ADL, and cognition QoL. The findings supported the olfactory transmission of α-synuclein to the cortices, substantia nigra.

## 1. Introduction

Parkinson’s disease (PD) is characterized by both motor and non-motor symptoms that have a great impact on quality of life (QoL). Dysosmia is one of the non-motor symptoms of PD, and its prevalence ranges from 46 to 97% [1,2]. Normosmia is a marker of younger onset with a PD phenotype that has a less severe motor manifestation [1]. Increasing evidence shows that dysosmia correlates with not only motor symptoms, but also various non-motor symptoms, including cognitive, psychiatric, sleep, and autonomic dysfunction [3,4,5]. 

A plausible pathomechanism underlying the correlation of dysosmia with motor and non-motor manifestation is that α-synuclein (αSN) spreads in a prion-like fashion from the olfactory bulb to other structures. It primarily spreads to the anterior olfactory nucleus [6], which connects to various locations, including the entorhinal cortices, olfactory tubercles, substantia nigra (SN), amygdala, dorsal motor nucleus of the vagus nerve, raphe nuclei, and locus coeruleus [7]. However, there is conflicting evidence regarding the correlation between dysosmia in PD with motor dysfunction [1,2,3,4,5,8,9,10,11,12] and depression [3,4,13], although there is a consistent correlation between dysosmia and cognitive dysfunction in PD [3,4,5,13,14]. A cohort study showed that only 51–83% of autopsies of PD patients followed the Braak staging, while another cohort showed that 6.3% deviated from the Braak staging but showed prominent involvement of olfactory structures and amygdala [15]. This reflects the need for more evidence exploring the role of dysosmia in the spreading of Lewy body pathology to the SN through the brainstem, limbic route, or cortical route [16,17]. 

Quality of life (QoL) reflects a patient’s holistic experiences of motor and non-motor symptoms. The 39-item Parkinson’s Disease Questionnaire (PDQ-39) is a thoroughly examined and widely utilized tool to assess QoL. It is used to measure severity levels in eight dimensions [18]. Additionally, part I and II of the Movement Disorder Society-Sponsored Revision of the Unified Parkinson’s Disease Rating Scale (MDS UPDRS) measure motor and non-motor experiences of activities of daily living. The correlation of dysosmia with QoL in PD had rarely been studied [19]. Therefore, in this cohort study, we aimed to evaluate the effect of dysosmia on QoL in PD. We hypothesized that αSN transmission through an olfactory route might cause more severe substantia-nigra symptoms, limbic symptoms, and cortical symptoms, and dysosmia might predict worse motor and non-motor QoL in patients with PD.

## 2. Materials and Methods

### 2.1. Study Design, Data Sources, and Population

The participants who met the International Parkinson and Movement disorder society Clinical Diagnostic Criteria for Parkinson’s disease [20] were recruited in the Taichung Veterans General Hospital from October 2016 to February 2021. Demographic variables, including age, age at onset, (AAO), disease duration, the Hoehn and Yahr scale, Levodopa equivalent daily dose (LEDD), and number of years of education, were recorded for all participants. PD severity was rated with the Movement disorder Society-sponsored revision of Unified Parkinson’s Disease Rating Scale (MDS UPDRS) [21].

Those who completed the assessments of the traditional Chinese version of the University of Pennsylvania Smell Identification Test (UPSIT) [10,22,23], the Montreal cognitive assessment (MoCA) [24], the MDS UPDRS [21], and the Chinese-translated version of the 39-item Parkinson’s Disease Questionnaire (PDQ-39) [25,26] were enrolled. The exclusion criteria were: upper respiratory tract infection during the test; documented nasal surgery; and failure to cooperate with the full clinical assessments. The present study was approved by the Taichung Veterans General Hospital Institutional Review Board and Ethics Committee (No. CE16171B). Written informed consents were provided by the participants before enrollment, in accordance with the ethical standards addressed in the Declaration of Helsinki.

### 2.2. Clinical Assessments

#### 2.2.1. Assessment of Olfactory Function

Each participant’s olfaction was assessed by the traditional Chinese version of the UPSIT (Sensonics, Inc., Haddon Heights, NJ, USA), which has been validated (Cronbach’s α = 0.637, test–retest reliability = 0.664) in the general population [23] and a PD population [10] in Taiwan. There were 40 odorants lodged in the “scratch and sniff” samples. After being properly instructed, the participants were requested to determine the correct odorant out of four options. One point was given if the correct odorant was chosen, and total points out of 40 were given to each participant. A cut-off value of 19 was applied to separate the total anosmia (i.e., UPSIT scored 0–18) and the non-anosmia groups (including the normosmia, mild; moderate; and severe microsmia) [10,23].

#### 2.2.2. Assessment of Disease Severity

The disease severity of the participants was evaluated with the MDS UPDRS in both “On” and “Off” status [21]. Briefly, part I comprises the non-motor elements of daily living (concerning cognition, psychiatric, sleep, autonomic disturbance, and pain), part II includes the motor experience of daily living, part III represents the motor examination assessed by the physicians, and part IV involves the treatment-related motor complications. The total score is the sum from part I to part IV. All items are scaled from 0 (normal) to 5 (severe). Among these items, part II 2.12–2.13 and part III 3.10–3.12 were utilized to assess the subjective experience and objective evaluation of posture and gait [27].

#### 2.2.3. Assessment of Quality of Life (QoL)

The Chinese-translated version of PDQ-39 was utilized to evaluate the QoL of the participants which has good reliability (Cronbach’s α = 0.58–0.96, test–retest reliability = 0.75–0.95) [25,26]. This questionnaire consists of 39 items, each scored from 0 (never) to 4 (always), with a higher score indicating more frequent or severe symptoms. Eight dimensions were measured, including the mobility (MOB, ten items), activities of daily living (ADL, six items), emotional well-being (EMO, six items), stigma (STI, four items), social support (SOC, three items), cognition (COG, four items), communication (COM, three items), and bodily discomfort (BOD, three items). The items of each dimension were averaged and standardized to give a score ranging from 0–100. The average of the eight dimensions gave a summary index (SI).

#### 2.2.4. Assessment of Cognitive Function

The Montreal Cognitive Assessment (MoCA) [24], a validated exam for assessing cognitive impairment, was employed to determine the global cognition in the study population. A cut-off value of 26 was used to indicate global cognitive dysfunction.

#### 2.2.5. Assessment of Dopamine Transporter Scan (DaTscan)

[2-[[2-[[[3-(4-chlorophenyl)-8-methyl-8-azabicyclo[3,2,1]-oct-2-yl]-methyl](2-mercaptoethyl)amino]ethyl]amino]ethanethiolato(3-)-N2,N2′,S2,S2′]oxo[1R-(exo-exo)]-[^99m^Tc]-technetium (Tc^99m^-Trodat-1) Single-Photon Emission Computed Tomography was utilized to assess Dopamine transporter (DaT) binding. A single bolus of 26.7 nCi of Tc^99m^-Trodat-1 was slowly injected intravenously to the patient. The DaTscan was obtained 3 h after the injection. A visual rating scale [28] was adopted to evaluate the extent of DaT binding (inter-rater agreement κ = 0.81). Briefly, 0 = bilateral normal striatal uptake; 1 = normal caudate uptake, putaminal uptake >50% on one side, and <50% on the other; 2 = normal caudate uptake, bilateral putaminal uptake <50%; 3 = caudate uptake <50%, no putaminal uptake; 4 = between 3 and 5; 5 = bilateral no striatal uptake.

Five nuclear radiologists executed the visual rating process. We retrospectively collected the data. We excluded (1) time from clinical evaluation to DaTscan > 1 year, (2) missing data, and (3) ratings that were not proposed in the original paper (i.e., 0–1, 1–2, 2–3, 3–4, 4–5).

### 2.3. Statistical Analysis

Statistical analysis was executed by MedCalc^®^ Statistical Software version 20.015 (MedCalc Software Ltd., Ostend, Belgium; https://www.medcalc.org (accessed on 30 November 2021)). The clinical characteristics, including the UPSIT, MoCA, MDS UPDRS, and PDQ-39 scores, DaTscan visual scale were tested for normality of distribution with the Kolmogorov–Smirnov test. The comparisons between the total anosmia and non-anosmia groups were examined with the Mann–Whitney U test, independent *t*-test, and chi-squared test for non-parametric continuous, parametric continuous, and categorical variables, respectively. The correlations between the MDS UPDRS and UPSIT scores, and between the PDQ-39 and UPSIT scores were examined by Spearman’s correlation. A linear regression of the UPSIT score with MDS UPDRS motor score, PDQ-39 ADL score, and DaTscan visual scale were executed. The multiple linear regressions were adopted to identify the significant predictors of MDS UPDRS motor score, PDQ-39 ADL score and DaTscan visual scale by using the enter method. To avoid collinearity of the covariate, the variance inflation factors (VIF) were also examined. We calculated forty minus UPSIT (the reverse UPSIT) to examine collinearity considering that the trends of motor dysfunction (MDS UPDRS part III increment) and age (increment) were opposite to dysosmia (UPSIT decrement). In the post hoc analyses, we analyzed the correlations between the UPSIT scores and the items within the PDQ-39 MOB, ADL, and COG dimensions, between the UPSIT scores and the MDS UPDRS items regarding posture and gait (part II 2.12–2.13; part III 3.10–3.12), and between the MoCA scores and the items within the PDQ-39 COG dimension. Two-tailed values of *p* < 0.05 were considered statistically significant.

## 3. Results

### 3.1. Patient Characteristics

The study examined 243 participants with PD who met both the inclusion and exclusion criteria. To deal with the incomplete data, we executed a partial deletion analysis and complete analysis, which did not alter the trend of the main results. Additionally, we performed a comparison between the data-missing group data-available group (Table A1). The participants’ age, age at onset (AAO), and UPSIT score followed a Gaussian distribution. The median age was 67 years, the median AAO was 62 years, the median disease duration was 30 months, 60% were male, and 19% were in Hoehn and Yahr (H&Y) stage 3–5. The median Levodopa equivalent daily dose (LEDD) was 375 mg, the median MoCA score was 26 points, the median education level was 9 years, the median DaTscan visual scale was 4, and the median UPSIT score was 16 points. In total, 146 (60%) and 97 (40%) patients were assigned to the total anosmia and non-anosmia groups, respectively. The total anosmia group had greater age (68 vs. 65 years, *p* = 0.0007), greater AAO (63 vs. 60 years, *p* = 0.0349), higher percentage with H&Y stage 3–5 (23% vs. 11%, *p* = 0.0169), and lower MoCA score (25 vs. 26, *p* = 0.0038) than the non-anosmia group. Meanwhile, they had comparable disease duration, sex distribution, LEDD use, education, and DaTscan visual scale (Table 1).

### 3.2. Correlation of MDS UPDRS and PDQ-39 with UPSIT

Regarding the MDS UPDRS scores, we reported the results for the “On” status rather than the “Off” status. Because of different pharmacokinetics in each individual, we could not predict the amount of dopaminergic medication remaining in the serum in the “Off” status. Only the part III scores followed a normal distribution. The total anosmia group had higher total (51 vs. 42, *p* = 0.0005), part II (9 vs. 7, *p* = 0.0067), and part III scores (32 vs. 26, *p* = 0.0005) than the non-anosmia group. Meanwhile, the subscale scores of non-motor aspects (part I) and motor complications (part IV) were similar. The PDQ-39 scores of the total anosmia group showed higher severity in the dimensions of activities of daily living (ADL; 13 vs. 4, *p* = 0.0047) and cognition (COG; 31 vs. 19, *p* = 0.0015) (Table 2).

In the correlation analysis, the trend was identical to the between-group analysis. With regard to the MDS UPDRS, the total score (*r* = −0.2368, *p* = 0.0003) and motor subscale scores (*r* = −0.1588, *p* = 0.0150 for part II; *r* = −0.2572, *p* = 0.0001 for part III) were mildly correlated with the UPSIT score. The ADL and COG dimensions of PDQ-39 were also mildly correlated with the UPSIT score (*r* = −0.1623, *p* = 0.0131 and *r* = −0.2229, *p* = 0.0006, respectively) (Table 3).

### 3.3. Regression Analysis of MDS UPDRS Part III and PDQ-39 ADL Severity and DaTscan Visual Scale

In the simple linear regression, the PDQ-39 ADL dimension did not follow a Gaussian distribution, so we separated the total sample by quartiles into four groups. The UPSIT score correlated with the MDS UPDRS part III score (*r*^2^ = 0.0661, *p* < 0.0001), PDQ-39 ADL quartile (*r*^2^ = 0.0231, *p* = 0.0202), and DaTscan visual scale (*r*^2^ = 0.0919, *p* = 0.0082) (Figure 1a–c).

In the multiple linear regression model, MDS UPDRS part III score was introduced as the dependent variable, and UPSIT score, age, sex, and disease duration as independent variables into the analysis. The UPSIT score (β = −0.3304, *p* = 0.0043) and age (β = 0.3164, *p* = 0.0004) were predictors of the MDS UPDRS part III score. We conducted the multiple linear regression analyses on PDQ-39 ADL quartile and DaTscan visual scale. The UPSIT score did not significantly correlate ADL impairment (β = −0.0139, *p* = 0.1688) but did correlate DaTscan visual scale (β = −0.0365, *p* = 0.0137). All variance inflation factors (VIF) were <10, and the reverse UPSIT score had the same VIF as the UPSIT score as a variable (Table 4).

### 3.4. Post Hoc Analyses

In the post hoc analyses, 1 out of 10 items of the mobility (MOB) dimension of PDQ-39 was correlated with the UPSIT score (carry shopping bags, *r* = −0.1920, *p* = 0.0037), whereas 2 out of 6 items of the activities of daily living (ADL) dimension were correlated with the UPSIT score (do buttons or shoe laces, *r* = −0.1393, *p* = 0.0360; cutting food, *r* = −0.1520, *p* = 0.0220) (Table 5).

Of the MDS UPDRS items regarding gait and posture (part II 2.12–2.13; part III 3.10–3.12), only freezing (part II 2.13) was correlated with the UPSIT score (*r* = −0.1398, *p* = 0.0345) (Table 6).

Three out of four items of the cognition (COG) dimension of PDQ-39 correlated with the UPSIT score (daytime sleepiness, *r* = −0.1537, *p* = 0.0205; concentration, *r* = −0.2033, *p* = 0.0021; and poor memory, *r* = −0.1934, *p* = 0.0034). These items also correlated with the MoCA score (daytime sleepiness, *r* = −0.2873, *p* < 0.0001; concentration, *r* = −0.2022, *p* = 0.0022; and poor memory *r* = −0.3075, *p* < 0.0001) (Table 7).

## 4. Discussion

In this study, we aimed to evaluate the effect of dysosmia on motor function and quality of life in patients with PD. We speculated that αSN transmission through an olfactory route might cause more severe motor, limbic, and cortical symptoms. Dysosmia may correlate worse motor and non-motor QoL in PD patients.

In the present study, 60% of PD patients had total anosmia (UPSIT score < 19), which is higher than the rate of 33–50% in other cohorts (using different translated versions of the UPSIT) [4,9,19]. The mean UPSIT score was 16 ± 8. This score was lower than in a previous Taiwanese cohort (mean 21 ± 7), which recruited patients with disease duration ≤ 2 years. The Taiwanese population in that study also scored 2.5–5 points lower on the UPSIT than North American norms [10], which might explain the relatively lower UPSIT score and frequent total anosmia in our cohort.

The total anosmia group had older age, higher AAO, higher disease severity, and more severe cognitive dysfunction than the non-anosmia group. The total anosmia group had a worse motor function according to the scores of MDS UPDRS part II and part III, and a worse QoL in the ADL and COG dimensions of PDQ-39. Meanwhile, non-motor scores of MDS UPDRS part I were comparable. The results of Spearman’s correlation showed the correlations of the UPSIT score with MDS UPDRS and PDQ-39 scores in the uniform dimensions. In the simple linear regression, dysosmia correlated with the MDS UPDRS part III score, ADL impairment, and DaTscan visual scale. In the multiple linear regression, dysosmia correlated the MDS UPDRS part III and DaTscan visual scale but not ADL impairment. Thus, the results showed that dysosmia correlated with motor dysfunction and worse QoL measurement of activities of daily living and cognition in PD patients and was indicative of a more severe phenotype. The results supported that dysosmia in PD is a marker of higher disease severity independent of disease duration [1,3] and supported the olfactory transmission of αSN to the SN and cortices.

Our results showed that dysosmia correlated well with motor dysfunction, which was evident in the MDS UPDRS part II, part III, and PDQ-39 ADL dimension. In the multiple regression analysis, dysosmia was a stronger predictor for the MDS UPDRS part III than ADL impairment, which might be explained by applying the ADL quartile instead of original ADL measures as a dependent variable. Although the link between dysosmia and motor dysfunction is supported by the connection between the olfactory tract and the SN [6,29,30], there have been contradictory results on the relationship between olfaction and motor function in PD patients. Studies have utilized different olfactory assessment tools, including odor threshold, detection, odor discrimination by the “Sniffin sticks” test [2,5,11,12], odor identification by different UPSIT versions [3,4,8,9,10], the 12-item version of UPSIT including brief-smell identification (B-SIT) [31], and the Cross-Cultural Smell Identification Test (CCSIT) [1]. In some studies applying the “Sniffin stick” test, the motor severity scores on the Unified Parkinson’s Disease Rating Scale (UPDRS) [32] motor subscale (part III) or Hoehn and Yahr stage were not correlated with olfactory function [2,11,12]. Meanwhile, in some of the studies using the UPSIT scores, the scores correlated with motor dysfunction assessed by the UPDRS part III [3,4,9], while some studies failed to show the correlation [8,10,19]. Different methodologies to assess general motor function [8], study population [19], and relatively mild disease severity [10] might also account for the lack of correlation between dysosmia and motor dysfunction.

Our results demonstrated that dysosmia correlated with the activities of daily living (ADL) dimension but did not correlate with the mobility (MOB) dimension. To clarify the discrepant correlation of ADL and MOB dimensions with dysosmia, we explored the detailed items in each dimension in post hoc analysis Compared to the items in the ADL dimension, items in the MOB dimension were more closely related to gait and postural instability. Only the item of “carry shopping bags” in MOB dimension had the correlation with higher UPSIT score (*r* = −0.1920, *p* = 0.0037). The post hoc analysis on the MDS UPDRS items regarding gait and posture (i.e., part II 2.12–2.13; part III 3.10–3.12) showed no significant correlation with dysosmia, except for “2.13 freezing” in part II. The results may indicate that dysosmia had less correlation with gait and postural instability in PD patients, which might account for the lack of correlation between dysosmia with MOB. Our results showed a general lack of correlation of postural and gait aspects with dysosmia. The postural instability and gait difficulty (PIGD) phenotype did not display a strong association with dysosmia in previous studies [31,33,34]. This might be explained by the involvement of postural and gait being relatively late in PD. Furthermore, the substantia nigra (SN) is closer to the olfactory route than the pedunculopontine nucleus (PPN), midbrain locomotor region (MLR), and other brainstem nuclei. Finally, Braak staging is best used in young-onset PD patients (age of onset: 55 ± 3 years) with long disease duration [35] but is not valid for all types of PD, implying that diverse manifestations occur in PD. However, studies showed that freezing of gait (FOG) occurs more often in PD with dysosmia [31,36] and that Parkinsonian gait progresses in the dysosmic elderly [37]. Therefore, further studies are required to explore the relationship between specific gait and postural disturbance with dysosmia in PD and other neurodegenerative diseases.

In the current study, the PDQ-39 cognition (COG) dimension showed correlations with dysosmia. The individual items within the COG dimension include the patient’s subjective experience of daytime sleepiness, concentration, poor memory, and distressing dreams and hallucinations. The four items correlated with dysosmia and MoCA except for “dreams and hallucinations”. The PDQ-39 COG dimension has been shown to correlate with individual neuropsychological tests [38], individual cognitive domains [39], and depression [39]. Our previous studies suggested there is a close relationship between dysosmia and impaired cognition according to neuropsychological testing [13,14], but there seemed to be a lack of correlation between dysosmia and depression [13,40]. The findings in our study revealed similar results. Dysosmia seemed to be correlated with the COG dimension of PDQ-39, but it lacked the association with the emotional well-being (EMO) dimension. “Daytime sleepiness” could represent various sleep disorders, such as sleep onset and maintenance insomnia, nocturnal restlessness, nocturnal motor symptoms, nocturia [41], or the effect of dopaminergic medication [42]. Excessive daytime sleepiness may also be exacerbated or caused by other secondary mechanisms such as obstructive sleep apnea or REM-sleep behavior disorder (RBD) [42]. It has been shown that daytime sleepiness correlates with PD dementia and more advanced disease [43]. Previous studies also showed a correlation between dysosmia and sleep disturbance [3,4], particularly daytime sleepiness [4]. However, in our study, there was no significant correlation between the UPSIT and MoCA scores with “dreams and hallucinations”, which indicated RBD or psychosis. The lack of link between dysosmia and distressing dreams and hallucinations could imply a brain-first PD process in patients with olfactory dysfunction, instead of a body-first PD process presenting RBD as a premotor symptom [17]. Jacob Horsager et al. [17] speculated that Parkinson’s disease comprises two subtypes: a brain-first type and a body-first type. In the “brain-first” type, the αSN pathology may enter via the olfactory bulb and spread to the brainstem and cortex. It results in marked involvement of the SN, moderate involvement of the pons, but little involvement of the medulla and autonomic nervous system (ANS). In contrast to a brain-first type, a body-first type is caused by the pathology originating in the enteric or peripheral ANS and then spreading to the brain. Due to less involvement of locus coeruleus from the olfactory route, RBD is less present in a brain-first (top-down) type than a body-first (bottom-up) type. It may explain why there was no significant correlation between dysosmia and the item of “dreams and hallucinations” in our results. However, this hypothesis requires more comprehensive studies to support it.

Although COG and ADL dimensions could have a prominent impact on a patient’s overall QoL, the summary index (SI) of PDQ-39 and the MDS UPDRS part I score failed to show a significant correlation with dysosmia. The PDQ-39 SI has identical weights of the eight dimensions, so outstanding dimensions could not be highlighted. MDS UPDRS part I encompasses a variety of non-motor symptoms but is non-specific. These non-motor symptoms include cognition, psychiatric, sleep, autonomic disturbance, and pain symptoms. Although anxiety, apathy, sleep, and autonomic disturbance have been correlated with dysomia [3,4,44], depression is still a matter of debate [3,4,13,40]. Furthermore, there is no pathophysiological basis for the relationship between pain and dysosmia at present.

To the best of our knowledge, there were very few studies discussing the relationship between olfactory dysfunction and QoL by using detailed dimensions of PDQ-39 in PD patients. We also correlated olfactory function with the items of the PDQ-39 regarding the mobility, activity of daily living, and cognition dimensions and tried to find the relationship. Besides, part I and II of MDS UPDRS, which also measure non-motor and motor experiences of activities of daily living, have not been correlated with dysosmia in previous studies. We applied these two assessment tools and analyzed the correlation between dysosmia and QoL in PD patients. In addition, we utilized DaTscan to indicate nigral dopaminergic dysfunction in the study.

This is an analytical cross-sectional study, and we collected data from a single medical center. There were several limitations. First, we did not use other assessment tools for nonmotor symptoms of PD, such as BDI-II for depression, PDSS-2 for sleep [41]. However, our primary aim was to study QoL in PD. PDQ-39 was recommended by the MDS task force to evaluate QoL in PD [18]. Additionally, PDQ-39 could serve as an indirect indicator of non-motor symptoms in PD [45]. Second, sleep and sexual dysfunction could have a significant impact on a patient’s QoL, but PDQ-39 is deficient in measurements in these dimensions. Third, DaTscans visual scale were only available in a subgroup of patients due to retrospective data collection. Hence, we performed a comparison between DaTscan visual rating-missing and available group (Table A1). Fourth, we were concerned about type I error when executing the post hoc analysis. However, the Bonferroni method was not utilized. We acknowledged that the questions within the same dimension were not independent, and its use was at the price of loss of power [46]. Fifth, we did not evaluate the type of hallucination along with the PDQ-39 item. Of note, olfactory hallucinations can occur in 11.3% of PD patients [47]. Finally, our study recruited patients from a single tertiary center with a median disease duration of 30 months, which might limit its generalizability.

## 5. Conclusions

In conclusion, our study provided clinical evidence that dysosmic patients with PD could have a significant motor and cognitive impairment affecting their QoL. The results revealed that dysosmia has a role in predicting more severe phenotypes affecting QoL independent of disease duration. Furthermore, our results supported the olfactory transmission of αSN to the cortices and SN and demonstrated that PD is a heterogeneous disorder with diverse clinical manifestations.

## Figures and Tables

**Figure 1 jpm-12-00754-f001:**
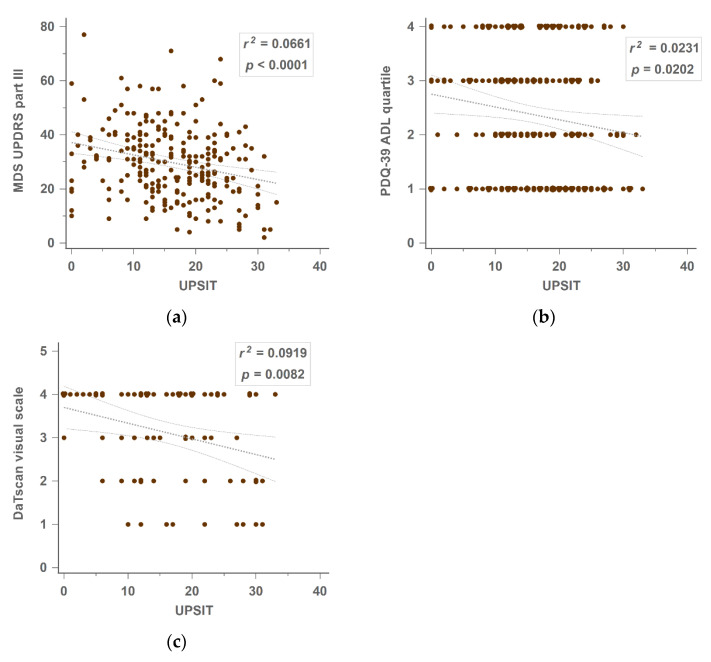
Linear regression of (**a**) MDS UPDRS part III on UPSIT, n = 234; (**b**) PDQ-39 ADL quartile on UPSIT, n = 233; (**c**) DaTscan visual scale on UPSIT, n = 75. ADL—activity of daily living; DaTscan—Tc^99m^ Dopamine transporter scan; MDS UPDRS—Movement Disorder Society-sponsored revision of the Unified Parkinson’s Disease Rating Scale; PDQ-39—39-item Parkinson’s Disease Questionnaire; UPSIT—traditional Chinese version of the University of Pennsylvania Smell Identification Test.

**Table 1 jpm-12-00754-t001:** Clinical features and comparison between the two groups.

	Total Sample	Total Anosmia	Non-Anosmia	*p* Value	n
		UPSIT < 19	UPSIT ≥ 19		
		n = 146	n = 97		
Age, years	67 (60–73)	68 (62–74)	65 (55–72)	0.0007 *	243
AAO, years	62 (54–70)	63 (57–70)	60 (53–68)	0.0349 *	241
Duration, months	30 (11–72)	36 (12–84)	24 (8–57)	0.0576	243
Male gender, n (%)	145 (60%)	90 (62%)	55 (57%)	0.4427	243
Hoehn and Yahr					234
0–2, n (%)	189 (81%)	106 (73%)	83 (86%)	0.0169 *	189
3–5, n (%)	45 (19%)	34 (23%)	11 (11%)		45
LEDD, mg	375 (150–680)	399 (160–749)	350 (118–608)	0.1335	242
MoCA	26 (21–28)	25 (20–28)	26 (24–29)	0.0038 *	243
Education, years	9 (6–14)	9 (6–14)	12 (6–14)	0.2514	243
DaTscan visual scale	4 (2–4)	4 (3–4)	3 (2–4)	0.0869	75
UPSIT	16 (11–22)	12 (8–15)	23 (21–26)		243

AAO—age at onset; DaTscan—Tc^99m^ Dopamine transporter scan; LEDD—Levodopa equivalent daily dose; MoCA—Montreal Cognitive Assessment; UPSIT—traditional Chinese version of the University of Pennsylvania Smell Identification Test. Continuous variables presented as median (interquartile ranges); categorical variables presented as number of patients (percentage). * *p* < 0.05.

**Table 2 jpm-12-00754-t002:** Comparison of MDS UPDRS and PDQ-39 between the two groups.

	Total Sample	Total Anosmia	Non-Anosmia	*p* Value
		UPSIT < 19	UPSIT ≥ 19	
MDS UPDRS	n = 234	n = 140	n = 94	
total	46 (34–64)	51 (38–69)	42 (29–53)	0.0005 *
part I	9 (5–14)	10 (5–15)	8 (5–11)	0.1067
part II	8 (3–14)	9 (4–16)	7 (3–12)	0.0067 *
part III	30 (20–39)	32 (23–40)	26 (16–33)	0.0005 *
part IV	0 (0–1)	0 (0–1)	0 (0–1)	0.7467
PDQ-39	n = 233	n = 141	n = 92	
SI	17 (9–28)	17 (9–30)	14 (7–27)	0.2124
MOB	15 (3–38)	15 (4–40)	13 (3–30)	0.2478
ADL	8 (0–25)	13 (0–25)	4 (0–21)	0.0047 *
EMO	13 (4–29)	13 (4–29)	17 (4–33)	0.9354
STI	6 (0–25)	6 (0–25)	6 (0–22)	0.9726
SOC	0 (0–42)	0 (0–35)	0 (0–42)	0.9727
COG	25 (13–44)	31 (17–50)	19 (6–38)	0.0015 *
COM	8 (0–25)	8 (0–25)	0 (0–21)	0.3175
BOD	17 (8–33)	17 (8–33)	25 (8–38)	0.2801

ADL—activities of daily living; BOD—bodily discomfort; COG—cognitions; COM—communication; EMO—emotional wellbeing; MDS UPDRS—Movement Disorder Society-sponsored revision of the Unified Parkinson’s Disease Rating Scale; MOB—mobility; PDQ-39—39-item Parkinson’s Disease Questionnaire; SI—summary index; SOC—social support; STI—stigma; UPSIT—traditional Chinese version of the University of Pennsylvania Smell Identification Test. Continuous variables presented as median (interquartile ranges). * *p* < 0.05.

**Table 3 jpm-12-00754-t003:** Correlation between MDS UPDRS and PDQ-39 with UPSIT.

	*r*	*p* Value	n
MDS UPDRS			234
total	−0.2368	0.0003 *	
part I	−0.1188	0.0696	
part II	−0.1588	0.0150 *	
part III	−0.2572	0.0001 *	
part IV	−0.0377	0.5669	
PDQ-39			233
SI	−0.0896	0.1730	
MOB	−0.0820	0.2127	
ADL	−0.1623	0.0131 *	
EMO	0.0281	0.6696	
STI	0.0324	0.6233	
SOC	−0.0249	0.7051	
COG	−0.2229	0.0006 *	
COM	−0.0645	0.3269	
BOD	0.0249	0.7058	

ADL—activities of daily living; BOD—bodily discomfort; COG—cognitions; COM—communication; EMO—emotional wellbeing; MDS UPDRS—Movement Disorder Society-sponsored revision of the Unified Parkinson’s Disease Rating Scale; MOB—mobility; PDQ-39—39-item Parkinson’s Disease Questionnaire; SI—summary index; SOC—social support; STI—stigma; UPSIT—traditional Chinese version of the University of Pennsylvania Smell Identification Test. * *p* < 0.05.

**Table 4 jpm-12-00754-t004:** Factors affecting MDS UPDRS part III, PDQ-39 ADL quartile, and DaTscan visual scale: multiple linear regression.

Variables	β	Std. Error	*r* _partial_	VIF	*p* Value	n
MDS UPDRS part III						234
UPSIT	−0.3304	0.1145	−0.1873	1.161	0.0043 *	
Age	0.3164	0.0884	0.2302	1.141	0.0004 *	
Sex	0.2758	1.7304	0.0105	1.019	0.8735	
Duration	0.0249	0.0152	0.1073	1.010	0.1037	
PDQ-39 ADL quartile						233
UPSIT	−0.0139	0.0101	−0.0910	1.134	0.1688	
Age	0.0157	0.0076	0.1352	1.116	0.0405 *	
Sex	−0.0149	0.1489	−0.0066	1.021	0.9202	
Duration	0.0065	0.0013	0.3195	1.005	<0.0001 *	
DaTscan visual scale						75
UPSIT	−0.0365	0.0144	−0.2894	1.296	0.0137 *	
Age	−0.0003	0.0120	−0.0028	1.168	0.9813	
Sex	0.0967	0.2502	0.0462	1.149	0.7003	
Duration	0.0019	0.0018	0.1249	1.030	0.2958	

ADL—activities of daily living; β—regression coefficient; DaTscan—Tc^99m^ Dopamine transporter scan; MDS UPDRS—Movement Disorder Society-sponsored revision of the Unified Parkinson’s Disease Rating Scale; PDQ-39—39-item Parkinson’s Disease Questionnaire; Std. Error—standard error; UPSIT—traditional Chinese version of the University of Pennsylvania Smell Identification Test; VIF—variance inflation factor. * *p* < 0.05.

**Table 5 jpm-12-00754-t005:** Correlation between PDQ-39 MOB and ADL dimension with UPSIT, n = 227.

Variables of PDQ-39	*r*	*p* Value
MOB		
PDQ1 Leisure activities	−0.0337	0.6140
PDQ2 Looking after home	−0.0581	0.3833
PDQ3 Carry shopping bags	−0.1920	0.0037 *
PDQ4 Walking half a mile	−0.0455	0.4950
PDQ5 Walking 100 yards	−0.0447	0.5028
PDQ6 Getting around the house	−0.0394	0.5547
PDQ7 Getting around in public	0.0614	0.3573
PDQ8 Need company when going	−0.1193	0.0729
PDQ9 Worry falling in public	−0.0373	0.5758
PDQ10 Confined to the house	0.0190	0.7764
ADL		
PDQ11 Washing	−0.1284	0.0533
PDQ12 Dressing	−0.1109	0.0955
PDQ13 Do buttons or shoelaces	−0.1393	0.0360 *
PDQ14 Writing clearly	−0.1091	0.1010
PDQ15 Cutting food	−0.1520	0.0220 *
PDQ16 Hold a drink without spilling	−0.1186	0.0745

ADL—activities of daily living; MOB—mobility; PDQ-39—39-item Parkinson’s Disease Questionnaire; UPSIT—traditional Chinese version of the University of Pennsylvania Smell Identification Test. * *p* < 0.05.

**Table 6 jpm-12-00754-t006:** Correlation between MDS UPDRS items regarding gait and posture with UPSIT, n = 229.

Variables of MDS UPDRS	*r*	*p* Value
part II		
2.12 Walking and balance	−0.0785	0.2366
2.13 Freezing	−0.1398	0.0345 *
part III		
3.10 Gait	−0.0891	0.1790
3.11 Freezing of gait	−0.0697	0.2935
3.12 Postural instability	−0.1112	0.0932

MDS UPDRS—Movement Disorder Society-sponsored revision of the Unified Parkinson’s Disease Rating Scale; UPSIT—traditional Chinese version of the University of Pennsylvania Smell Identification Test. * *p* < 0.05.

**Table 7 jpm-12-00754-t007:** Correlation between PDQ-39 COG dimension with UPSIT and MoCA, n = 227.

Variables of PDQ-39	UPSIT		MoCA	
	*r*	*p* Value	*r*	*p* Value
COG				
PDQ30 Daytime sleepiness	−0.1537	0.0205 *	−0.2873	<0.0001 *
PDQ31 Concentration	−0.2033	0.0021 *	−0.2022	0.0022 *
PDQ32 Poor memory	−0.1934	0.0034 *	−0.3075	<0.0001 *
PDQ33 Dreams and Hallucinations	−0.1026	0.1231	−0.0972	0.1445

COG—cognitions; MoCA—Montreal Cognitive Assessment; PDQ-39—39-item Parkinson’s Disease Questionnaire; UPSIT—traditional Chinese version of the University of Pennsylvania Smell Identification Test. * *p* < 0.05.

## Data Availability

The data presented in this study are available on request from the corresponding author. The data are not publicly available due to privacy issues.

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
