# Peer review of "Dysosmia Is a Predictor of Motor Function and Quality of Life in Patients with Parkinson’s Disease"

_jpm, 2022, doi:10.3390/jpm12050754_

Round 1

Reviewer 1 Report

This paper is well written and judged to be appropriate.
And it will provide useful information to clinicians.
However, as the authors mentioned in the limitation section, it is unfortunate that there is no comparison with the DAT PET scan.
Since the DAT PET scan is routinely used in the recent diagnosis of Parkinson's disease, it is thought that the quality of the paper will be much improved compared to the various imaging parameters obtained here. If there is a DAT scan, it would be good to add it. If this is not possible, please describe this in more detail in the limitation section.

I thank the authors for their hard work.

Author Response

Point 1:  If there is a DAT scan, it would be good to add it.

Response 1: Thank you for your valuable suggestion. Thus, we added DaT SPECT in our analysis.

line 23 (abstract),

line 126 to 140; 146-147; 153-155 (methods),

line 184; Table1; Figure1c; line 240-241 245-247; line 257-259; Table4; line 264(results),

line 328-331; 427-428; 436-438 (discussion),

Table A1 (appendix).

Reviewer 2 Report

The Manuscript ID: jpm-1704312, titled “Dysosmia is a predictor of motor function and quality of life in 2 patients with Parkinson's disease" aimed to examine the effect of dysosmia on motor function and QoL in PD. The article is methodologically correct, well organized and elegantly proposed. It is a bit long-winded but still interesting and easy to read.

Specific comments:

At line 18 it should be changed the term "predicted" with "correlates"
At line 70-75 if you have references of validation of chinese version of UPSIT and PDQ-39 tests, please provide it
At line 84 you forgot suffix "chinese version" of UPSIT ?
At line 149-150 you indicates that all 223 participants met both the inclusion criteria and the exclusion criteria, but this would exclude the totality of the subjects from the study group
At line 288 it should be changed the term "predicted" with "correlates"
In the table 7 Authors indicate correlations between  PDQ-34 Dreams and hallucinations with UPSIT and MoCA without any description of type of hallucinations. Probably it consists on visual hallucination but it is not clear to understand.

Line 341 Authors should describe the type of  hallucinations since a previous study (Solla et al., 2020, https://doi.org/10.3390/
brainsci11070841) indicated that  olfactory hallucinations  occur frequently in PD patients, especially in women, and
often concomitant with visual and auditory hallucinations without any association with olfactory impairment.

Author Response

Point 1:

At line 18 it should be changed the term "predicted" with "correlates"

At line 288 it should be changed the term "predicted" with "correlates"

Response 1: Thank you for your valuable suggestions.

Hence, we changed the term to correlate (line21 and 314).

Point 2:

At line 70-75 if you have references of validation of chinese version of UPSIT and PDQ-39 tests, please provide it

Response 2: Thank you for the friendly reminder.

Hence, we added the reference of Chinese version of PDQ-39 (reference #26) in line 78. We also mentioned the validated Chinese version of UPSIT and PDQ-39 in line 90-92, line 111-112

Point 3:

At line 84 you forgot suffix "chinese version" of UPSIT ?

Response 3: Thank you for the friendly reminder.

We added the suffix in line 89.

Point 4:

At line 149-150 you indicates that all 223 participants met both the inclusion criteria and the exclusion criteria, but this would exclude the totality of the subjects from the study group

Response 2: We appreciate your valuable suggestion.

Hence, we executed a complete anaysis with 243 patients with PD. This did not change the main result of the study. We also performed a comparison between data-missing and data-available group in the appendix. In addition, we added DaT SPECT in our study (suggestion from reviewer 1).

line 18-23(abstract),

line 169-184 ; Table1 and line 188; line 202-206; Table2; line219-222; Table3; line 234-241; figure1a-1c and line245-249; line 253-262; Table4 and line 264; line 275-280; Table5; line 288-291; Table6; line 298-303; Table7. (results),

line 317; 322; 328-331; 361 (discussion),

Table A1 (appendix).

Point 5:

In the table 7 Authors indicate correlations between  PDQ-34 Dreams and hallucinations with UPSIT and MoCA without any description of type of hallucinations. Probably it consists on visual hallucination but it is not clear to understand.

Response 5: Thank you for the valuable suggestion.

However, we did not assess the specific type of hallucinations. Hence, we put this in our limitations in line 441-444.